# Healthcare resource utilisation pattern and costs associated with herpes simplex virus diagnosis and management: a systematic review

Shaun Wen Huey Lee [1,2,3] Sami L Gottlieb,[4] Nathorn Chaiyakunapruk[1,5,6]

[1]School of Pharmacy, Monash University Malaysia, Bandar Sunway, Selangor, Malaysia
[2]School of Pharmacy, Taylor's University Lakeside Campus, Subang Jaya, Malaysia
[3]Center of Global Health, University of Pennsylvania, Philadelphia, Pennsylvania, USA
[4]Department of Reproductive Health and Research, World Health Organization, Geneve, Switzerland
[5]Department of Pharmacotherapy, University of Utah, Salt Lake City, Utah, USA
[6]School of Pharmacy, University of Wisconsin, Madison, WI, USA

**Correspondence to**
Dr Nathorn Chaiyakunapruk; nathorn.chaiyakunapruk@utah.edu

## ABSTRACT

**Objectives** Little is known about the economic burden of herpes simplex virus (HSV) across countries. This article aims to summarise existing evidence on estimates of costs and healthcare resource utilisation associated with genital and neonatal HSV infection.

**Design** Systematic literature review.

**Data sources** Seven databases were searched from inception to 31 August 2020. A focused search was performed to supplement the results.

**Eligibility criteria** Studies which reported either healthcare resource utilisation or costs associated with HSV-related healthcare, including screening, diagnosis and treatment of genital HSV infection and neonatal herpes prevention and treatment.

**Data extraction and synthesis** Two independent reviewers extracted data and assessed the risk of bias using the Larg and Moss's checklist. All data were summarised narratively.

**Results** Out of 11 443 articles, 38 were included. Most studies (35/38, 94.6%) were conducted in high-income countries, primarily the United States, and were more often related to the prevention or management of neonatal herpes (n=21) than HSV genital ulcer disease (n=17). Most analyses were conducted before 2010. There was substantial heterogeneity in the reporting of HSV-related healthcare resource utilisation, with 74%–93% individuals who sought care for HSV, 11.6%–68.4% individuals who received care, while neonates with herpes required a median of 6–34 hospitalisation days. The costs reported were similarly heterogeneous, with wide variation in methodology, assumptions and outcome measures between studies. Cost for screening ranged from US$7–100, treatment ranged from US$0.53–35 for an episodic therapy, US$240–2580 yearly for suppressive therapy, while hospitalisation for neonatal care ranged from US$5321–32 683.

**Conclusions** A paucity of evidence exists on healthcare resource utilisation and costs associated with HSV infection, especially among low-income and middle-income countries. Future research is needed on costs and healthcare utilisation patterns to improve overall understanding of the global economic burden of HSV.

## INTRODUCTION

Herpes simplex virus (HSV)-1 and HSV-2 are DNA viruses that belong to Alphaherpesviridae, a subfamily of the Herpesviridae family.[1] Both

### Strengths and limitations of this study

► This is the first systematic review to assess the healthcare resource utilisation and costs associated with herpes simplex virus (HSV) infections.
► Comprehensive literature searches were conducted, which were supplemented by a focused search.
► Heterogeneity of study designs and outcome measures limited the meta-analysis of study results.
► Relatively few studies described the healthcare resource utilisation patterns and cost of HSV, especially from low–middle income countries.

viruses can cause genital infection, which can have a profound impact on sexual and reproductive health. HSV-2 is almost entirely transmitted during sexual activity and is the most common cause of genital herpes, affecting more than one in every 8 individuals, or 491.5 million people, aged 15–49 years in 2016.[2] HSV-1 is the main cause of oral herpes but can also be transmitted to the genital area through oral sex. HSV-1 affects an estimated 3.7 billion people under age 50 globally, of which over 120 million may have genital infection.[2] While the prevalence of HSV infection is high globally, it varies widely by region. The highest prevalence of both HSV-1 (88% in women and men) and HSV-2 (44% in women; 25% in men) is in the African region, which is primarily comprised of low-income and middle-income countries (LMICs).[1 2]

Genital HSV infection is lifelong and characterised by periodic reactivation. Many infections are asymptomatic or unrecognised, but up to a third of people may develop painful, recurrent genital sores known collectively as genital ulcer disease (GUD).[3] Antiviral medications can be taken episodically to shorten GUD outbreaks or taken daily (suppressive therapy) to reduce the number of outbreaks, but they are not curative. Pregnant women with genital HSV infection can also transmit the virus to their infants in the peripartum

period, resulting in neonatal herpes.[4] Although this occurs only rarely, neonatal herpes has a high fatality and disability rate among surviving infants. As such, particularly in high-income countries (HICs), prevention measures such as caesarean section are often undertaken if a mother has active HSV lesions at delivery. Genital HSV-2 infection has also been linked to an increased risk of acquisition and transmission of human immunodeficiency virus (HIV) infection.[5]

The World Health Organisation (WHO) has highlighted the need for a vaccine against HSV-2, due to large numbers of infections globally and the resulting disease consequences including GUD, neonatal herpes, and increased risk of HIV acquisition.[6–8] Multiple vaccine candidates have been studied to date with modelling studies showing that prevention of HSV-2 infection with a vaccine could potentially also reduce the incidence of HIV infection.[9] Vaccines targeting HSV-2 might also have benefits against HSV-1.[10] Understanding the potential value of HSV vaccines requires not only predicting the impact of the vaccines on HSV-related disease burden, but also on its economic burden. However, little is known about the economic burden of HSV globally. As a first step in estimating HSV-related economic burden, we conducted a broad systematic review with the aim of summarising all available evidence on costs and resource utilisation associated with diagnosing, treating and managing genital and neonatal HSV infection.

## METHODS

The current study followed the guidelines of the *Cochrane Handbook for Systematic Reviews of Intervention*.[11] The review was reported in accordance with the Preferred Reporting Items for Systematic Reviews and Meta-Analyses.[12]

### Data sources and search strategy

We electronically searched for relevant articles published from database inception to 31 August 2020 in seven databases: PubMed, PsychINFO, Embase, Centre for Review and Dissemination, EconLit, CEA registry and WHO Library Database. The search strategy was based on a broad combined search string "Herpes Simplex Virus" AND "cost" OR "resource utilization" OR "econ*", with no language restriction. A complete search strategy is detailed in online supplemental appendix text 1. In addition, bibliographies of relevant articles were examined to identify potential studies not indexed in the aforementioned databases. A focused supplemental search on Google Scholar was performed using the keywords listed in online supplemental appendix text 2 based on the inclusion above.

### Study selection

Studies were included if they were original articles that investigated resource utilisation patterns and costs related to HSV infection including the cost of any diagnostic tools, consultation time, treatment and hospital cost related to detecting and managing all types of HSV-1 or HSV-2 related neonatal and genital infections and associated disease outcomes. We included articles which were published in English languages.

### Data extraction and quality assessment

The study followed a 2-stage process, where two independent reviewers screened the titles and abstracts for relevant studies, before the full texts were screened by another two independent reviewers for eligibility. Relevant information from the identified studies was extracted independently by two reviewers using a standardised data extraction sheet. At all stages, any disagreement was resolved by discussion between reviewers through consensus. Information collected from the data extraction sheet included: (1) general study information including country of the study, (2) HSV subtype and disease, (3) study design, (4) healthcare resource utilisation, (5) costs of relevant tests, clinical care, hospitalisation and medications and (6) summary estimates of HSV-related economic burden. Methodological quality of all included economic studies was assessed using the Consensus Health Economic Criteria list. This checklist has been recommended for critically appraising published economic evaluations. The checklist has 19 domains and includes reporting standards for economic model characteristics (population, time horizon, perspective and discount rate), identification and valuation of costs and outcomes, discussion points, conclusions as well as funding and conflicts of interest. All cost of illness studies were evaluated for risk of bias using the Larg and Moss's checklist. No quality appraisal was performed on studies reporting healthcare resource utilisation.

### Data analysis

A component-based analysis was used to describe and synthesise the overall findings from all included studies. Specifically, tabulation methods were used to report on study characteristics, outcomes and costs. Tables for resource utilisation and disaggregated costs were presented and summarised. All costs were presented according to the recommendations of Turner *et al*.[13] For studies that did not provide the year of cost data, the year of publication was used. Adjustment for inflation was done using the Gross Domestic Product deflator (GDP deflator) of the studied country. Cost estimates were then converted and reported in 2017 United States Dollars (USD). GDP deflator and exchange rates were obtained from the World Bank.[14]

### Patient and public involvement

Patients were not involved in this systematic review. Their input was not sought in the design, interpretation or writing of the document.

## RESULTS

### Study selection

Our search yielded a total of 11 443 articles of which 8779 articles were excluded as they were not relevant for this review based on title screening. The remaining 2664 articles were further screened by title and abstract and 299

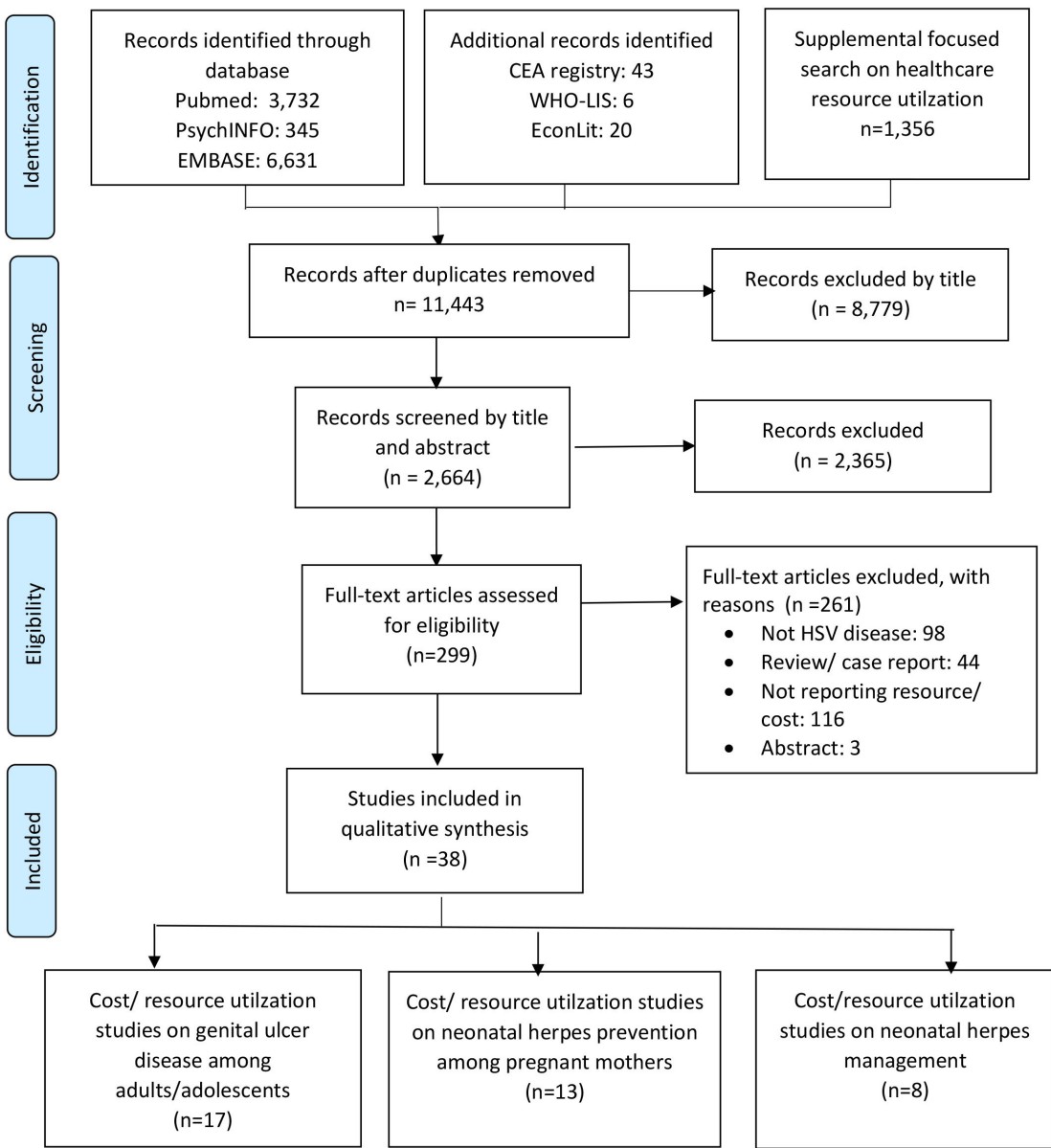

**Figure 1** Flow diagram of study selection process. HSV, herpes simplex virus.

articles were assessed for inclusion. We excluded 261 articles (n=98 for not related to HSV, n=44 review articles/case report, n=116 not reporting resource utilisation or cost, n=3 available only in abstract), leaving a total of 38 studies included in this review, as shown in figure 1.

### Overview of study characteristics

Of the 38 included articles, 14 studies[15–28] described resource utilisation only, 12 studies[29–40] reported on costs and 12 studies[41–52] reported both resource utilisation and costs of HSV diagnosis/management. These studies, published from 1989 to 2020, reported resource utilisation or costs related to the diagnosis and management of HSV-related GUD among adults/adolescents[18–22 28 30–34 37–40 44 52] (n=17), neonatal herpes prevention in pregnant mothers (n=13)[23–25 27 29 35 36 42 43 46–49] and neonatal herpes management[15–17 26 41 45 50 51] (n=8). The majority of studies were conducted in high income countries (HIC) (35/38, 94.6%) including the United States (USA)[15 17 20 22 25 27 29 30 34 35 38–52] (n=26), Canada[18 19 26 36] (n=4), United Kingdom (UK)[23 33] (n=2), France[16 28] (n=2) and Ireland[24] (n=1)), while only one study (1/38, 2.6%) was conducted in a middle-income country, in particular South Africa.[32] A global survey focusing on the experiences of patients receiving care for genital herpes in 78 countries included some data on healthcare utilisation.[21] In addition, a modelling study estimated the costs of implementing the Global Health Sector Strategy on Sexually Transmitted Infections (STIs), 2016–2021, in 117 LMICs, including costs related to syndromic management of GUD, the vast majority of which is caused by HSV-2.[37] The quality of included studies is summarised in online supplemental appendix figures 1 and 2.

## Methodological heterogeneity

There was substantial heterogeneity in the reporting of the included studies. Most studies were cost or resource utilisation studies (n=23), while the remaining were cost-effectiveness studies (n=15). Among cost or resource utilisation studies, data were collected retrospectively (n=13), prospectively (n=7) or not reported (n=7). The number of participants in each study varied, which could be as few as 39 participants to as large as 42 million in studies that analysed claims datasets. Twenty-one studies (21/38, 55.3%) included participants who had either HSV-1 or 2, 10 studies (10/37, 27.0%) specifically included participants with HSV-2, while the remaining eight studies (8/38, 21.1%) did not specify which type of HSV they examined. A summary of the characteristics of these studies is presented in online supplemental appendix table 1, and study findings are presented in online supplemental appendix tables 1 and 2 (see appendix for detailed unit cost tables and accompanying references).

## Cost and healthcare resource utilisation pattern of genital herpes infection

Among all 17 studies[18–22 28 30–34 37–40 44 52] investigating cost and healthcare resource utilisation pattern of genital herpes, 11 studies reported some cost components of care for genital herpes infection[30–34 37–40 44 52] (online supplemental appendix tables 1, 2 and 4). All but one of these studies were conducted in HIC and only one LMIC study (from South Africa) was found. The cost components of the included studies were variably reported. Three studies[31 34 52] reported laboratory testing costs associated with diagnosing HSV. Eight studies[30 31 33 34 37 40 44 52] described costs associated with syndromic management of GUD. In four studies,[32 33 37 52] the authors describe the drug charges associated with treatment or prevention of HSV using oral acyclovir (doses of 200–400 mg). The cost reported varied considerably, ranging between US$0.53 and US$16 for a 5–7 day treatment course for episodic GUD and US$40 for a month of suppressive therapy with acyclovir. Two studies[31 44] provided the total drug charges associated with overall management of GUD, but no details related to the treatment regimen, duration or HSV of HSV being treated (online supplemental appendix table 2). Seven studies[31–33 37 47 48 52] described labour and service delivery costs such as cost of physician visits, drug procurement cost, counselling cost and clinical examination associated with HSV. Similarly, there was variation in terms of reported labour and service delivery cost, which could be as low as US$0.28 for 10-min counselling[33] to as high as US$120 for consultation and lost wages of patient time.[52] Indirect costs were considered only by Szucs et al who estimated HSV-related productivity losses, which was estimated at a US$60 visit.[31]

Considering the cost components together, Owusu-Edusei et al estimated that the lifetime direct medical cost per case of genital HSV infection in the USA (considering only GUD-related costs and adjusted to 2017 USD) was US$855 among men (range: US$428–$1284) and US$698 among women (range: US$350–1047).[30] This translated to a total cost of US$607.3 million (range: US$303.59–910.89 million in 2017 USD) for lifetime management of new or newly diagnosed cases of HSV-2 in the USA occurring in 2008. Scuzs et al meanwhile estimated that the annual direct and indirect medical costs in the USA would amount to US$983 million, based on an estimated 3.1 million symptomatic genital HSV episodes (both new and recurrent) a year.[31]

The only middle-income country study, from South Africa,[32] reported the diagnostic/operational costs associated with medication, staff and laboratory costs for daily HSV-2 suppressive therapy among people living with HIV.[32] The median cost for HSV-2 suppressive therapy per life-year gained ranged between US$685 and US$951 (adjusted to 2017 dollar) among HIV-1 infected antiretroviral naïve women. The authors estimated that this could be a cost-effective method for delaying HIV disease progression, especially when the price of acyclovir was lower than the price of US$0.026 per day for a two times per day 400 mg dose. However, this study was conducted when antiretroviral therapy (ART) use was recommended only when CD4 count fell below a threshold of <200 cells/µL or <350 cell/µL (online supplemental appendix table 5). On a more global level, in Korenromp et al's cost estimates for implementing the Global STI Strategy in 117 LMIC over 2016–2021, the authors reported that it would cost approximately US$109 million to diagnose and treat HSV-related GUD episodes seen in clinical care, not including service delivery costs.[37] These costs were estimated despite assuming that only about 4% of all HSV-2 infected people would seek care for GUD (15% recognising symptoms and 28% of those seeking care).

A total of eight studies described healthcare resource utilisation patterns for genital herpes infection,[18–22 31 40 44] and all were from high-income countries (online supplemental appendix tables 1 and 3). Five of these studies[18 20–22 40] reported the population rate of seeking medical care for HSV, based on retrospective analyses of databases of patients from health surveys.[20–22] In the study by Di Xia et al, the authors found that the total genital herpes associated emergency department (ED) use increased from 24 747 visits in 2006 to 36 518 in 2013.[40] It is important to note that none of the studies reported the proportion of those seeking medical care among HSV-infected individuals. Most of these consultations were relatively short in nature, and were less than 15 min (79%).[21] Two studies described the diagnostic methods used to determine HSV among their population. In the first study conducted in 2004, Patrick et al surveyed physicians in 78 countries and reported that the most commonly used test was viral culture, which was performed in 49% of the individuals[21] (online supplemental appendix table 3). A recent study in France by Heggarty et al in 2020 found that 43.3% of respondents in their survey stated that they would conduct polymerase chain reaction (PCR) test plus HSV serology and another

39.9% would conduct PCR test only to confirm a HSV diagnosis.[28]

Treatment patterns of individuals with genital herpes were also reported in four studies.[19 21 28 44] The study by DesHarnais et al in 1996 reported on antiviral use only among hospitalised patients with herpes infections, which is unlikely to be representative of the vast majority of people with HSV infection. Patrick et al in their survey found that 65% of people with genital herpes had ever been treated with antivirals, while 18% used topical prescription medication and 13% used over the counter topical cream. Among these individuals, 67% had received episodic therapy while 31% received chronic suppressive therapy (online supplemental appendix table 2). Another study on herpes-related quality of life reported that 76.9% of respondents had ever been treated with antivirals, and 33.3% of the respondents with HSV were on suppressive antiviral therapy when the survey was administered.[19]

## Cost and healthcare resource utilisation pattern of prevention of neonatal herpes among pregnant mothers

Nine studies reported costs for neonatal herpes prevention among pregnant mothers[29 35 36 42 43 46–49] (online supplemental appendix tables 1, 2 and 6). Seven studies[35 36 42 43 46 47 49] provided estimates on the cost for treatment and childbirth delivery options, including caesarean and vaginal delivery in addition to inpatient costs. The cost of hospitalisation ranged considerably, and could be as low as US$300 to as high as US$32 483, while the cost of delivery ranged between US$2300 and US$9490. The costs associated with different laboratory tests used, such as ELISA screening or viral cultures[36 43] were reported, while detailed listing of the cost component of different delivery methods and hospital care were included in some studies (online supplemental appendix table 6). The cost-effectiveness studies examined the impact of either acyclovir suppressive therapy[29 35 46 47] or routine antenatal screening[36 42 43 48 49] for prevention of neonatal herpes. In a study by Randolph et al,[47] the authors found that prophylaxis with acyclovir during late pregnancy could be a cost-effective strategy to reduce the need for caesarean delivery due to genital herpes outbreaks during labour. Baker et al further expanded this work and estimated that adding serological testing to antiviral suppressive therapy had an incremental cost per quality-adjusted life year gained of US$18 680, compared with no screening or suppressive therapy.[42] A modelling study by Tuite et al had similar findings related to screening for HSV in pregnancy.[36]

Our focused search found a total of 10 studies which reported resource utilisation among pregnant mothers to prevent neonatal herpes.[23–28 42 43 46 48] Among these, four were cost-effectiveness studies which had provided some information regarding resource utilisation based on estimates from literature or assumptions.[42 43 46 48] In one of the earliest studies by Brocklehurst in 1995, a survey of British obstetrician–gynaecologists revealed that most would recommend some form of antenatal screening for

HSV using viral cultures usually by week 34 of gestation.[23] However, such screening is no longer recommended in the UK. Studies within HICs that have national obstetrics guidelines recommending caesarean delivery when HSV lesions are present at delivery have shown that most clinicians follow this guidance.[24–27] For example, in a Canadian study, caesarean section was offered 'most of the time' to women with HSV lesions at delivery by 92% of obstetricians and 82% of family physicians.[26] In addition, in these settings women with genital herpes are often offered antiviral suppressive therapy in the third trimester.[24 26] Both valacyclovir and acyclovir have been used, with difference in preference by country. In the most recent survey of clinicians managing pregnant women with HSV by Heggarty et al in 2020, the authors noted that 68.4% 'always' prescribe suppressive antiviral therapy during the third trimester and an additional 11.6% 'often' prescribe it for women with symptomatic primary HSV infection during pregnancy.[25] For women with recurrent symptoms during pregnancy, 55.1% of providers always prescribe and 12.9% often prescribe antiviral prophylaxis in the third trimester.[28]

## Cost and healthcare resource utilisation pattern of neonatal herpes management

Four studies[41 45 50 51] reported cost of neonatal herpes management and reported only direct medical costs (online supplemental appendix tables 1 and 2). One study reported direct non-medical cost for long-term care of individuals with neurological disability due to sequelae of HSV.[43] All studies were in HIC. The reported cost of hospitalisation of neonatal HSV ranged considerably, from US$27 843 to US$92 664. One study reported the cost associated with hospital readmission, which was reportedly similar to the first hospitalisation episode.[50] Six studies[36 46–49 52] accounted for the costs of informal care in their calculation. Informal caregiving was defined as care provided by caregivers for infants who had neurological sequelae following neonatal herpes. In total, seven studies[36 43 46–49 52] estimated long-term care costs of neonatal herpes patients. One of these, by Thung and Grobman,[49] provided the estimated cost for long-term care of neonates with mild neurological deficit due to HSV, which cost US$17 304.61 after adjusting for inflation to 2017 values. Six studies[43 46–49 52] provided estimates for the lifetime cost of caring for a child with moderate and severe disability, and fall within the range US$68 894–US$432 263 and US$232 698–US$ 1 296 792, respectively. It is important to note that all studies relied on estimation of long-term costs calculated by Weitzman et al[53] with some different assumptions, while one study[43] used other sources of data.

A total of seven studies[15–17 41 45 50 51] described resource utilisation among individuals with neonatal herpes (online supplemental appendix tables 1 and 3). These studies described the length of stay for hospitalisation which varied considerably, with median hospital stays ranging from 6 to 34 days[15 16] Ahmad et al noted that

nearly 9.4%–9.8% of neonates who had HSV required ICU stay.[15] None of the studies reported the number of days for intensive care unit (ICU) hospitalisation.

## DISCUSSION

Our review revealed a heterogeneous body of evidence on the healthcare resource utilisation and costs associated with genital and neonatal HSV infection, as well as some summary economic estimates and cost-effectiveness studies of HSV intervention strategies, such as use of antivirals or screening, which included unit cost data. While the evidence base provides a starting point for understanding, several gaps remain. Despite the broad search strategy and inclusion criteria, we identified only 38 papers, which shows the paucity of data on HSV-related healthcare resource utilisation as well as economic costs, especially from LMIC settings. The lack of data from LMIC is particularly concerning, as these countries bear the greatest burden of HSV infection and disease.[2 3 54] The current review only identified one cost-effectiveness analysis from a middle-income country[32] focused on people living with HIV only, and one high-level modelling study predicting costs of implementing care for HSV GUD across 117 LMIC globally.[37] In addition, many of the studies we found were relatively old and may not reflect current practices such as the use of newer diagnostics (eg, PCR test) and newer care recommendations. For example, the global study by Patrick et al reported that viral culture was the most common test used to diagnose HSV but this is likely because the use of PCR test was not yet common in clinical practice at the time of the study. The 2020 study in France by Heggarty et al reveals that PCR test is now the most commonly used test, at least in this HIC setting, with and without HSV serology.[28]

While data on resource utilisation and costing were most comprehensive from the USA, large gaps remain in many areas. For example, Gilbert et al[20] described the proportions of individuals seeking care for genital herpes among adults aged 18–24 from 2000 to 2006, but since then there have been no new updates. In terms of costing, we noticed similar trends, as studies[30] mostly referenced cost data collected in 2001 by Szucs et al.[31] This lack of data is similarly noted related to HSV infection during pregnancy. While some information from health surveys exists, healthcare resource utilisation information is rarely tracked or reported. Our search demonstrated that for most of the world, data on HSV related resource utilisation are sparse. As such, new data sources and better data collection efforts are needed to collect these standardised non-fatal data from diverse healthcare settings. One major need is an understanding of how closely clinicians follow national guidelines on HSV care and treatment, such as the studies by Kenny et al[26] and Heggarty et al[28] from Canada and France, respectively. For example, while there are structured guidelines for the workup of neonatal herpes and its related management, our review did not identify any studies that described the compliance

to these guidelines. Such information can provide us with vital clues into the economic burden of neonatal HSV as there is substantial cost due to the high mortality rates neonatal HSV was not treated.

Our review was also constrained in summarising findings across studies or countries and in conducting across-study comparisons, due to the limited data and differing methodologies, healthcare settings, and practices, particularly for healthcare resource utilisation. Another concern was the heterogeneity in data presentation in many studies identified. For example, the length of hospital stay reported in studies varied considerably, with different assumptions used by authors, and as a result, the cost of hospitalisation varied significantly even within the USA, which limits the potential generalisability of these findings across different settings.[16 41 45 51] Healthcare practices also differ between LMIC and HIC with respect to how HSV is managed, for example, most HSV cases in LMICs are treated as part of syndromic management for GUD, without diagnostic testing. This may mean that additional testing costs might need to be considered for HICs, whereas additional treatment, for example, for syphilis and chancroid, which can also cause GUD syndromes, might need to be considered for LMICs. The focus on GUD more generally in LMICs may have made it more challenging to identify potentially relevant HSV-specific studies for LMIC settings.

In order to estimate the global economic burden of HSV to contribute to the understanding of the potential value of HSV interventions, research on HSV-related costs and healthcare utilisation patterns is urgently needed, especially from LMIC settings. Standardisation of methods for the measurement and reporting of economic costs would enhance across-study comparisons and inform prioritisation strategies of global funders. Only one study broadly attempted to quantify the economic burden of HSV, which the authors estimated would require a projected investment of around US$109 million from 2016 to 2021, just for the management of HSV-associated GUD, not considering service delivery costs.[37] However, this analysis only modelled treatment of HSV GUD for a small proportion of people with HSV-2 infection (approximately 4%; assuming 15% would recognise symptoms and 28% of those would seek care) and did not account for HSV recurrences within a given year. New global estimates of HSV GUD suggest this is likely an underestimate.[3] In addition, as this model lacked country-level estimates of baseline disease and did not take into account the full spectrum of disease outcomes related to HSV nor the burden on health systems, the costing estimates remain imprecise and incomplete, suggesting the need for a more comprehensive model.

This is the first systematic review of scientific literature on the healthcare resource utilisation for HSV. We conducted a comprehensive literature search and included grey literature through our focused search. Nevertheless, most studies were only conducted in HIC especially from the USA. As the practice and thus utilisation of resources

will vary between settings and countries due to epidemiological and health systems differences, this will limit the generalisability of findings. Nevertheless, results of this study will serve as a future repository for studies that wish to examine the economic evaluations of any public health interventions for HSV. This review also highlights the importance and need for more studies to describe on the healthcare resource utilisation and associated cost of HSV, especially from LMIC. We assessed study quality of all included studies, which allows readers to assess the internal validity of these studies. The literature search was also limited to studies published in English language. As data on healthcare resource utilisation may be published in government reports, or book chapters, these may not have been retrieved and included in this review, which may partly explain the lack of studies describing healthcare resource utilisation from LMIC.

## CONCLUSION

This review is the first attempt and a key step towards providing data needed to understand the global economic burden of HSV infection, for both HICs and LMICs. Available economic estimates, primarily from HICs, suggest the economic burden of HSV infection could be substantial. However, the global picture remains incomplete. Nevertheless, results obtained from this study will form a repository which can inform future economic evaluations of interventions for HSV infection, including HSV vaccines, microbicides or new antiviral medications.[55] These types of economic data are crucial not only to improve the planning and development of any future HSV-related healthcare interventions, but also to optimise the allocation of healthcare expenditures and medical resources.

**Contributors** SWHL served as the lead author, conducted the research, conducted the analyses, integrated the input from all team members and drafted the initial manuscript. SLG directed the initial research and contributed to the initial draft, integrated her view points and served as an expert in this work. NC conducted the research, mediated the discussion and helped refine the draft. All authors approved the final manuscript. NC is the guarantor and accepts full responsibility for the work and/or the conduct of the study, had access to the data, and controlled the decision to publish

**Funding** This work was funded by the WHO Department of Sexual and Reproductive Health and Research, via support from the UNDP-UNFPA-UNICEF-WHO-World Bank Special Programme of Research, Development and Research Training in Human Reproduction (HRP) and the U.S. National Institute of Allergy and Infectious Diseases, part of the National Institutes of Health (U01 AI108543).

**Competing interests** None declared.

**Patient consent for publication** Not applicable.

**Ethics approval** This study does not involve human participants.

**Provenance and peer review** Not commissioned; externally peer reviewed.

**Data availability statement** All data relevant to the study are included in the article or uploaded as supplementary information.

**ORCID iD**
Shaun Wen Huey Lee http://orcid.org/0000-0001-7361-6576

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
