## [Reviewer comments · BMJ Open]

ARTICLE DETAILS

TITLE (PROVISIONAL)	Healthcare resource utilization pattern and costs associated with herpes simplex virus diagnosis and management : a systematic review
AUTHORS	Lee, Shaun Wen Huey; Gottlieb, Sami L.; Chaiyakunapruk, Nathorn

VERSION 1 – REVIEW

REVIEWER	Chaabane, Sonia University of Montreal
REVIEW RETURNED	12-Mar-2021

GENERAL COMMENTS	This systematic review summarizes existing evidence on estimates of costs and resource utilization associated with diagnosing, treating, and managing HSV infection and disease, and specific cost drivers across healthcare systems. The review revealed a heterogeneous body of evidence and a paucity of data on the health resource utilization and costs associated with genital and neonatal HSV infection, as well as summary economic estimates. Limited data was available from LMIC settings. While the evidence base provides a starting point for understanding the HSV disease burden several gaps remain. Main limitations of this review: 1) The lack of data from LMIC countries (bear the greatest burden of HSV infection and disease) is particularly concerning as relevant data sources for LMIC were not searched. 2) The mixed study designs, populations, countries, study period creates heterogeneous body of evidence and leading to inclusive results. Abstract The objective in the abstract does not reflect the specific outcomes of interest in the systematic review. What is the meaning of methodological variation in healthcare systems? is this a research outcome? The results section of the abstract focuses on methodological differences rather than the results. There is a need to present and discuss the main results for each outcome and sub-population. Introduction
---

	Page 5, line 19. Would be more informative to add the prevalence (%) of both HSV-1 and HSV-2 in the African region. Page 5, line 43, HIV needs to be defined on the first-time use. Page 5, line 47, add the citation to WHO report/ data highlighting the need for a vaccine against HSV-2. The objective of the systematic review needs to be specific and align throughout the article. Methods Did the systematic review followed any standard guidelines for conducting systematic reviews (e.g. Cochrane)? The PRISMA checklist for reporting systematic reviews was used along with the PRISMA flowchart. Please mention that in the methods section. Is there an a priori registered protocols for this systematic review? Results On page 11, lines 53-60 “ At the time of the study, the use of PCR was not yet common in clinical practice. A recent study in France by Heggarty et al. in 2020 found that PCR is now more commonly used, with 43.3% of respondents in their survey stated that they would conduct PCR in addition to HSV serology while another 39.9% would conduct PCR only to confirm a HSV diagnosis” . I suggest to add this point to the discussion section as it is not part of the main outcomes. In the paragraph of the results on the cost and health resource utilization pattern of prevention of neonatal herpes among pregnant mothers, on page 12, it would be informative to report the cost ranges by the country income status (LMIC, HIC). This factor could explain the wide variability of the observed cost ranges. Not sure how cost-effectiveness analysis and high-level modeling prediction studies answer the research questions of interest in this review as the main objective of these study designs is to compare the costs and effectiveness of 2 interventions or predict the effect of an intervention, respectively. The country specificity (income, access and availability to health resources during the study period) and the period of data collection/analysis have an important impact on the studied outcomes. Many of the included studies are relatively old and reflect different practices and may not reflect current practices such as the use of newer diagnostics and care. Conscious of the limited number of available studies, I recommend to report and discuss the results based on a specific and justified time threshold. Discussion Most of the data from LMIC and Asian countries are published in grey literature and local sources and not indexed in the searched databases. This is the most likely explanation to the limited data
--	---

	found from LMIC. This should be highlighted in the limitations of your review. In page 16, line 51, it is mentioned that grey literature was included. Nowhere in the methods section, it has been mentioned that you search grey literature sources. If so, this should be clearly mentioned in the methods section and the used grey literature sources should be listed. In page 16, line 60. It is mentioned that “the literature search was also limited to English language”. However, in the methods it is mentioned that “the search strategy was based on a broad combined search string “Herpes Simplex Virus” AND “cost” OR “resource utilization” OR “econ*”, with no language restriction”. Please clarify.
--	---

REVIEWER	Weber, Zachary Walter Reed National Military Medical Center, Neonatology
REVIEW RETURNED	05-Aug-2021

GENERAL COMMENTS	This is a well written article and I think adds to the literature concerning resource utilization which is becoming of increasing interest. The authors do a great job reviewing the current published practices. It is not surprising how varied the costs are even within developed countries. I do feel the following areas could enhance the article. 1) neonatal workup and management is very protocolized. While it is not needed to go over the full work up in this article, I believe it is important to mention due to the work up and treatment while waiting for results is a substantial cost due to the high mortality rates when not treated. 2) PCR is becoming increasingly utilized in work up and diagnosis for HSV in all age ranges. There have been recent studies showing PCR results can decrease hospital duration and treatment time when a result is negative especially in pediatric and neonatal patients. I can see how some of these articles were left out with the study selection, but use of new technologies would be of benefit in this paper.
---

VERSION 1 – AUTHOR RESPONSE

Reviewer# 1 had the following 20 comments

Comment #1. This systematic review summarizes existing evidence on estimates of costs and resource utilization associated with diagnosing, treating, and managing HSV infection and disease, and specific cost drivers across healthcare systems. The review revealed a heterogeneous body of evidence and a paucity of data on the health resource utilization and costs associated with genital and neonatal HSV infection, as well as summary economic estimates. Limited data was available from LMIC settings. While the evidence base provides a starting point for understanding the HSV disease burden several gaps remain.

We thank the reviewer for highlighting the importance of this study.

Comment #2 The lack of data from LMIC countries (bear the greatest burden of HSV infection and disease) is particularly concerning as relevant data sources for LMIC were not searched.

We take note. As highlighted by reviewer, our systematic review only found a limited number of studies from LMIC. To ensure that we have not missed any of these relevant studies, we also performed a focused search to supplement our initial search. Unfortunately, we did not find any studies that had described on the healthcare resource utilisation from LMIC except from South Africa and none from a low-income country. These limitations have been highlighted in our discussion section as follow

Nevertheless, most studies were only conducted in HIC especially from the USA. As the practice and thus utilization of resources will vary between settings and countries due to epidemiological and health systems differences, this will limit the generalisability of findings. Nevertheless, results of this study will serve as a future repository for studies that wish to examine the economic evaluations of any public health interventions for HSV. This review also highlights the importance and need for more studies to describe on the healthcare resource utilization and associated cost of HSV, especially from LMIC. We assessed study quality of all included studies, which allows readers to assess the internal validity of these studies. The literature search was also limited to studies published in English language. As data on healthcare resource utilization may be published in government reports, or book chapters, these may not have been retrieved and included in this review, which may explain the lack of studies describing healthcare resource utilization from LMIC.

Comment #3. The mixed study designs, populations, countries, study period creates heterogeneous body of evidence and leading to inclusive results.

We agree. We have included these important points into our revised discussion section. The revised discussion now reads as follows

Nevertheless, results of this study will serve as a future repository for studies that wish to examine the economic evaluations of any public health interventions for HSV.

Comment #4. The objective in the abstract does not reflect the specific outcomes of interest in the systematic review. What is the meaning of methodological variation in healthcare systems? is this a research outcome?

This was unclear in our initial submission. We have revised our abstract objective to clarify that the aim of the current review as to summarise the existing studies that have reported on the healthcare resource utilisation and costs for HSV. The revised abstract reads as follow

Objectives: The World Health Organization (WHO) highlights the need for a vaccine against herpes simplex virus (HSV), in part due to the high disease burden globally. However, little is known about the economic burden of HSV across countries. This article aims to summarize existing evidence on estimates of costs and healthcare resource utilization associated with genital and neonatal HSV infection.

Comment #5. The results section of the abstract focuses on methodological differences rather than the results. There is a need to present and discuss the main results for each outcome and sub-population.

We thank the reviewer for the suggestion. We have revised our results section to discuss the main results for each outcome as suggested.

Comment #6. Page 5, line 19. Would be more informative to add the prevalence (%) of both HSV-1 and HSV-2 in the African region.

We thank the reviewer for the suggestion. The revised sentence now includes the prevalence of infection and reads as follows:

While the prevalence of HSV infection is high globally, it varies widely by region. The highest prevalence of both HSV-1 (88% in females and males) and HSV-2 (44% in females; 25% in males) is in the African region, which is primarily comprised of low- and middle-income countries (LMIC).^{1 2}

Comment #7. Page 5, line 43, HIV needs to be defined on the first-time use.

We take note and apologise for the oversight. We have expanded the term HIV as suggested before using any abbreviations in the article. We have also cross-checked the entire manuscript to ensure we define any abbreviations first

Comment #8. Page 5, line 47, add the citation to WHO report/ data highlighting the need for a vaccine against HSV-2.

We take note and included 3 additional citations as suggested that highlighted the need for a vaccine. The references are as follow

- Gottlieb SL, Low N, Newman LM, et al. Toward global prevention of sexually transmitted infections (STIs): the need for STI vaccines. *Vaccine* 2014;32(14):1527-35.
- Gottlieb SL, Giersing BK, Hickling J, et al. Meeting report: Initial World Health Organization consultation on herpes simplex virus (HSV) vaccine preferred product characteristics, March 2017. *Vaccine* 2019;37(50):7408-18.
- Gottlieb SL, Deal CD, Giersing B, et al. The global roadmap for advancing development of vaccines against sexually transmitted infections: Update and next steps. *Vaccine* 2016;34(26):2939-47. doi: <https://doi.org/10.1016/j.vaccine.2016.03.111>

Comment #9 The objective of the systematic review needs to be specific and align throughout the article.

We take note of the comment. As described in our introduction, the current work is part of the WHO's effort in developing a HSV vaccine for global use. As such, there is a need to understand the economic value and its potential impact if such vaccine was developed. As a first step, we plan to develop an economic model to quantify the burden of HSV. However, this is not possible without a better understanding of the current healthcare resource utilisation and associated costs related to HSV treatment or prevention. Thus, this systematic review was performed to quantify and summarise all related evidence on the cost and healthcare resource utilisation associated with HSV diagnosis, treatment and management. We have made further edits to our introduction to clarify this point and ensured that the aims and objectives in both abstract and the article are congruent as follow:-

The World Health Organization (WHO) has highlighted the need for a vaccine against HSV-2, due to large numbers of infections globally and the resulting disease consequences including GUD, neonatal herpes, and increased risk of HIV acquisition.(3-5) Multiple vaccine candidates have been studied to

date with modelling studies showing that prevention of HSV-2 infection with a vaccine could potentially also reduce the incidence of HIV infection.(6) Vaccines targeting HSV-2 might also have benefits against HSV-1.(7) Understanding the potential value of HSV vaccines requires not only predicting the impact of the vaccines on HSV-related disease burden, but also on its economic burden. However, little is known about the economic burden of HSV globally. As a first step in estimating HSV-related economic burden, we conducted a broad systematic review with the aim of summarizing all available evidence on costs and resource utilization associated with diagnosing, treating, and managing genital and neonatal HSV infection.

Comment #10, Did the systematic review followed any standard guidelines for conducting systematic reviews (e.g. Cochrane)?

Thank you very much for the suggestion. We have revised our methods to state the following

The current study followed the guidelines of the Cochrane Handbook for Systematic Reviews of Intervention.(8) The review was reported in accordance with the Preferred Reporting Items for Systematic Review and Meta-Analyses.(9)

Comment #11. The PRISMA checklist for reporting systematic reviews was used along with the PRISMA flowchart. Please mention that in the methods section.

Thank you very much for the suggestion. We have revised our methods to state the following

The current study followed the guidelines of the Cochrane Handbook for Systematic Reviews of Intervention.(8) The review was reported in accordance with the Preferred Reporting Items for Systematic Review and Meta-Analyses.(9)

Comment #12. Is there an a priori registered protocols for this systematic review?

We did not register the current review in PROSPERO due to time constraints. We note that this may not be the most ideal practice as suggested by Cochrane.

Comment #13. On page 11, lines 53-60 “ At the time of the study, the use of PCR was not yet common in clinical practice. A recent study in France by Heggarty et al. in 2020 found that PCR is now more commonly used, with 43.3% of respondents in their survey stated that they would conduct PCR in addition to HSV serology while another 39.9% would conduct PCR only to confirm a HSV diagnosis” . I suggest to add this point to the discussion section as it is not part of the main outcomes.

We thank the reviewer for the suggestion. We have now edited the manuscript and moved the following paragraph to the discussion as suggested. Our revised discussion now reads as follow

In addition, many of the studies we found were relatively old and may not reflect current practices such as the use of newer diagnostics (e.g. PCR) and newer care recommendations. For example, the global study by Patrick *et al.* reported that viral culture was the most common test used to diagnose HSV but this is likely because the use of PCR was not yet common in clinical practice at the time of the study. The 2020 study in France by Heggarty *et al.* reveals that PCR is now the most commonly used test, at least in this HIC setting, with and without HSV serology²⁸.

Comment #14. In the paragraph of the results on the cost and health resource utilization pattern of prevention of neonatal herpes among pregnant mothers, on page 12, it would be

informative to report the cost ranges by the country income status (LMIC, HIC). This factor could explain the wide variability of the observed cost ranges.

We thank the reviewer for the suggestion to improve our manuscript clarity. As described in our results, there was only one study from LMIC. All the cost reported for healthcare resource utilisation pattern for neonatal herpes were from high-income countries, and in particular the United States. This is reported in our discussion as follows:

Our review was also constrained in summarizing findings across studies or countries and in conducting across-study comparisons, due to the limited data and differing methodologies, healthcare settings, and practices, particularly for healthcare resource utilization. Another concern was the heterogeneity in data presentation in many studies identified. For example, the length of hospital stay reported in studies varied considerably, with different assumptions used by authors, and as a result, the cost of hospitalisation varied significantly even within the United States, which limits the potential generalizability of these findings across different settings

Comment #15. Not sure how cost-effectiveness analysis and high-level modeling prediction studies answer the research questions of interest in this review as the main objective of these study designs is to compare the costs and effectiveness of 2 interventions or predict the effect of an intervention, respectively.

As described in our review objective, our aim was to summarize all available evidence on the costs and resource utilisation associated with diagnosis, treatment or management of genital and neonatal HSV. The cost-effectiveness analysis and modelling studies provided us with some data on costs or healthcare resource utilisation, including in the parameters used for modelling, and thus met the inclusion/exclusion criteria of our study.

Comment #16. The country specificity (income, access and availability to health resources during the study period) and the period of data collection/analysis have an important impact on the studied outcomes. Many of the included studies are relatively old and reflect different practices and may not reflect current practices such as the use of newer diagnostics and care.

We take note. Unfortunately, as described in our discussion section, there was only limited data published on healthcare resource utilization worldwide related to HSV which highlights the need to report this information. This is described as follows

Our review revealed a heterogeneous body of evidence on the health resource utilization and costs associated with genital and neonatal HSV infection, as well as some summary economic estimates and cost-effectiveness studies of HSV intervention strategies, such as use of antivirals or screening, which included unit cost data. While the evidence base provides a starting point for understanding, several gaps remain. Despite the broad search strategy and inclusion criteria, we identified only 38 papers, which shows the paucity of data on HSV-related healthcare resource utilization as well as economic costs, especially from LMIC settings. The lack of data from LMIC is particularly concerning, as these countries bear the greatest burden of HSV infection and disease

We have also included a statement in our study limitation to describe the need to publish such studies in the future

This review also highlights the importance and need for more studies to describe on the healthcare resource utilisation and associated cost of HSV, especially from LMIC.

Comment #17. Conscious of the limited number of available studies, I recommend to report and discuss the results based on a specific and justified time threshold.

We take note and thank the reviewer for the suggestion.

Comment #18. Most of the data from LMIC and Asian countries are published in grey literature and local sources and not indexed in the searched databases. This is the most likely explanation to the limited data found from LMIC. This should be highlighted in the limitations of your review.

We thank the reviewer for the suggestion. We have included this in the study limitation as follow

The literature search was also limited to studies published in English language. As data on healthcare resource utilization may be published in government reports, or book chapters, these may not have been retrieved and included into this review, which may partly explain the lack of studies describing healthcare resource utilisation from LMIC.

Comment #19. In page 16, line 51, it is mentioned that grey literature was included. Nowhere in the methods section, it has been mentioned that you search grey literature sources. If so, this should be clearly mentioned in the methods section and the used grey literature sources should be listed.

We thank the reviewer for highlighting this important point. We have edited the methods under Data Sources and Search Strategy as follow:-

We electronically searched for relevant articles published from database inception to August 31st 2020 in 7 databases: PubMed, PsychINFO, EMBASE, Centre for Review and Dissemination, EconLit, CEA registry and WHO Library Database (WHOLIS). The search strategy was based on a broad combined search string "Herpes Simplex Virus" AND "cost" OR "resource utilization" OR "econ*", with no language restriction. A complete search strategy is detailed in Appendix 1. In addition, bibliographies of relevant articles were examined to identify potential studies not indexed in the aforementioned databases. A focused supplemental search on Google Scholar was performed using the keywords listed in Appendix 2 based upon the inclusion above.

Comment #20. In page 16, line 60. It is mentioned that "the literature search was also limited to English language". However, in the methods it is mentioned that "the search strategy was based on a broad combined search string "Herpes Simplex Virus" AND "cost" OR "resource utilization" OR "econ*", with no language restriction". Please clarify.

We apologise for the confusion. In our search, we did not restrict the search to only English language, but in our study selection, only English language articles were included. This is clarified in our revised methods, under study selection as follows:

Studies were included if they were original articles that investigated resource utilization patterns and costs related to HSV infection including the cost of any diagnostic tools, consultation time, treatment and hospital cost related to detecting and managing all types of HSV-1 or HSV-2 related neonatal and genital infections and associated disease outcomes. We included articles which were published in English languages.

Reviewer# 2 had the following 3 comments

Comment #1. This is a well written article and I think adds to the literature concerning resource utilization which is becoming of increasing interest. The authors do a great job reviewing the current published practices. It is not surprising how varied the costs are even within developed countries.

We thank the reviewer for the encouraging comments.

Comment #2. Neonatal workup and management is very protocolized. While it is not needed to go over the full work up in this article, I believe it is important to mention due to the work up and treatment while waiting for results is a substantial cost due to the high mortality rates when not treated.

We thank the reviewer for the suggestion to improve our manuscript. We have included this into our revised manuscript under discussion to state the following:-

One major need is an understanding of how closely clinicians follow national guidelines on HSV care and treatment, such as the studies by Kenny et al(10) and Heggarty et(11) al from Canada and France respectively. For example, while there are structured guidelines for the workup of neonatal herpes and its related management, our review did not identify any studies that described the compliance to these guidelines. Such information can provide us with vital clues into the economic burden of neonatal HSV as there is substantial cost due to the high mortality rates neonatal HSV was not treated.

Comment #3. PCR is becoming increasingly utilized in work up and diagnosis for HSV in all age ranges. There have been recent studies showing PCR results can decrease hospital duration and treatment time when a result is negative especially in pediatric and neonatal patients. I can see how some of these articles were left out with the study selection, but use of new technologies would be of benefit in this paper.

We take note and thank the reviewer for pointing this very important observation. As the reviewer pointed out, these studies were possibly not identified due to the inclusion/exclusion criteria of our study. This is highlighted in our revised discussion under study limitation as follows:

In addition, many of the studies we found were relatively old and may not reflect current practices such as the use of newer diagnostics (e.g. PCR) and newer care recommendations. For example, the global study by Patrick *et al.* reported that viral culture was the most common test used to diagnose HSV but this is likely because the use of PCR was not yet common in clinical practice at the time of the study. The 2020 study in France by Heggarty *et al.* reveals that PCR is now the most commonly used test, at least in this HIC setting, with and without HSV serology²⁸.

We trust that these changes will address the concerns of both reviewers and editors. We also wish to thank both reviewers and editor for their invaluable time taken to review our manuscript and providing us with the comments which significantly improved our manuscript.

VERSION 2 – REVIEW

REVIEWER	Weber, Zachary Walter Reed National Military Medical Center, Neonatology
REVIEW RETURNED	04-Oct-2021

GENERAL COMMENTS	Again, this is a well written article enhanced by the edits and additions made to the manuscript suggested by the reviewers. The authors carefully reviewed and appropriately responded to all the reviewers' comments and suggestions. The thorough explanation of the exclusion of more recent studies exploring more recent technologies and LMIC data is much improved and well explained in the limitations. This is sufficient due to the the strict study selection. This systemic review will be an important source, especially as cost and resource utilization analyses are of increasing interest.
--